# Novel Therapeutic Approaches Targeting Post-Translational Modifications in Lung Cancer

**DOI:** 10.3390/pharmaceutics15010206

**Published:** 2023-01-06

**Authors:** Maria Francesca Baietti, Raj Nayan Sewduth

**Affiliations:** 1TRACE, Laboratory for RNA Cancer Biology, Department of Oncology, KU Leuven, Herestraat 49, 3000 Leuven, Belgium; 2VIB-KU Leuven Center for Cancer Biology, Herestraat 49, 3000 Leuven, Belgium

**Keywords:** lung cancer, drug delivery, adenovirus, PROTAC, LNP, post-translational modification

## Abstract

Lung cancer is one of the most common cancers worldwide. It consists of two different subtypes: non-small cell lung cancer (NSCLC) and small cell lung cancer (SCLC). Despite novel therapeutic options such as immunotherapy, only 20% of lung cancer patients survive the disease after five years. This low survival rate is due to acquired drug resistance and severe off-target effects caused by currently used therapies. Identification and development of novel and targeted therapeutic approaches are urgently required to improve the standard of care for lung cancer patients. Here, we describe the recent development of novel drug-delivery approaches, such as adenovirus, lipid nanoparticles, and PROTACs, that have been tested in clinical trials and experimentally in the context of fundamental research. These different options show that it is now possible to target protein kinases, phosphatases, ubiquitin ligases, or protein modifications directly in lung cancer to block disease progression. Furthermore, the recent acceptance of RNA vaccines using lipid nanoparticles has further revealed therapeutic options that could be combined with chemo-/immunotherapies to improve current lung cancer therapies. This review aims to compare recent advances in the pharmaceutical research field for the development of technologies targeting post-translational modifications or protein modifiers involved in the tumorigenesis of lung cancer.

## 1. Introduction

In this review, we describe novel therapeutic approaches that take advantage of protein modifications in lung cancer. Protein modifications include phosphorylation, or the addition of a phosphorylated residue on a serine, threonine, or tyrosine, and ubiquitination, addition of a ubiquitin molecule to a lysine residue. While ubiquitination is involved in protein degradation through the proteasome and trafficking to vesicles, phosphorylation is important for the (in)activation of proteins, especially in signaling cascades such as mitogen activated protein kinase (MAPK) signaling cascade that is involved in cancer cell proliferation and survival. Kinases and phosphatases are enzymes involved in the phosphorylation or removal of the phosphorylation from a protein cargo. Several phosphorylation cascades are important for tumorigenic transformation or progression and have been extensively characterized. Several therapeutic design options focusing on specific kinases or phosphatases dysregulated in lung cancer progression were developed in the past years with various levels of success. This is key for the field as kinases, phosphatases, or ligases affecting oncogenic signaling are often mutated or deleted in lung cancer.

In the case of kinases and phosphatases, these therapeutic strategies involved adenoviral-based or lipid nanoparticle-based drug delivery approaches. In the case of ubiquitination, the E3 ubiquitin ligases and E2 ubiquitin conjugating enzymes are the proteins that define the ubiquitination reaction specificity and catalyze the ubiquitination, respectively. Several E3 ligases have been described to act as tumor suppressors or promoters, further linking this protein modification with oncogenesis. Recently of interest in the drug development field, taking advantage of the ubiquitin system, proteolysis targeting chimeras (PROTACs) were shown to be promising approaches in order to trigger the degradation of oncoproteins, while deubiquitinase targeting chimeras (DUBTACs) allowed the stabilization of onco-suppressors. PROTACs and DUBTACs are specific probes that recruit ubiquitin ligases, induce degradative ubiquitination, or ubiquitin hydrolases (de-ubiquitinases) that remove degradative ubiquitin chains from specific targets.

These different innovative approaches show that it is now possible to target protein modifications directly to block tumor progression. Targeting tumor specific protein modifications could also reduce off-target effects of anticancer drugs and side effects. Other protein modifications occur frequently in cancer, such as oxidation, glycosylation, nitrosylation, or ISGylation. These modifications directly affect the function of oncoproteins and oncosuppressors but have not been characterized enough yet, making it impossible to design therapies targeting them yet.

## 2. Adenoviral Targeting of Oncogenic Protein Modifications

Adenoviruses, many of which are responsible for respiratory infections in humans, have been genetically engineered to allow the delivery of genetic material to cells. These viruses demonstrate the ability to target a various range of cell types, as it is possible to use different serotypes that have specific affinity to specific cell types. Adenoviruses have a strong affinity for lung epithelial cells, from which most lung cancers are driven. Adenoviral-based therapies are genetic therapies that enable the expression of a protein of interest at a high level without showing any DNA integration or immune response towards the recombinant adenovirus. The efficiency of such therapies has been demonstrated in several genetic diseases (i.e., Zolgensma) and, more recently, by the use of adenoviral-based vaccines against COVID-19 respiratory diseases. Several clinical trials have tested with some degree of success adenovirus-based approaches to impair oncogenesis, based on specific cancer-specific genetic mutations such as the one affecting p53 function, as indicated by Table 1 [1,2,3,4,5,6,7,8,9]. On the other hand, clinical trials focusing on the immune response, such as anti-dendritic cell or cytokine therapy, were less successful but demonstrated an interesting proof-of-concept approach (Table 1, [3,10]). The large number of clinical trials using adenoviral-based approaches indicates the important potential of adenoviral-based therapies for lung cancer, especially considering the recent acceptance of the adenovirus-based vaccines in the general population with the COVID-19 vaccines. Further optimizations are still required to reach the full potential of adenoviral-based therapies in cancer patients.

Recent studies indicate that the primary receptor of adenoviruses on the cell surface, Coxsackie-adenovirus receptor (CAR), is expressed at different levels in various types of tumors. This could result in resistance to adenovirus infection in some cancer types, indicating that the adenoviral approach could be suboptimal in some types of cancer. However, a recent publication showed that the Coxsackie-adenovirus receptor is expressed at a high level in non-small cell lung cancer [14], indicating that adenoviral-based therapies should be very efficient in lung cancer [15], when compared to other cell types. The authors also showed that CAR expression was directly correlated with the epithelial-mesenchymal transition, a process important for lung tumor transformation and that is also correlated with tumor aggressiveness and resistance to therapy. This indicates that the most aggressive tumor cells would be the ones most infected using adenoviral-based approaches, further strengthening the potential of adenoviral-based therapies in the context of lung cancer resistant to classical therapies. It is still essential to assess the effects of adenoviral particles when delivered over time, as long-term anti-cancer treatment with such viruses could present side effects.

Furthermore, several adenoviral-based clinical trials indicated above in Table 1 [1,2,3,4,5,6,7,8,9,11] showed side effects, including fever, hypoxia, and dyspnea. Some more severe toxicities were observed sporadically in some trials, such as autoimmune diseases, cardiological alterations, local or systemic allergies, or, in extreme cases, organ failure. These effects were mostly observed in the case of anti-immune cells or anti-cytokine therapy (Table 1, [3,9,12]). This explains why most of these phase I/II clinical trials were not followed by a phase III/IV trial. Therefore, it is important to optimize adenoviral-based therapies to minimize side effects. It is now possible to target adenoviruses to a specific cell population using specific promoters controlling the expression of the adenoviral coding sequence. For example, in the case of lung adenocarcinoma, it would be possible to target specifically lung alveolar cells using the SFTPC (surfactant protein-C) promoter or CEBPα (CCAAT/enhancer-binding protein alpha) [16], dramatically reducing off-target effects when compared to classically used constitutive promoters. Finally, to target tumor angiogenesis to block neo-vessel formation and impair tumor progression, studies have shown that it is possible to target proliferative vascular endothelium using the ROBO4 (Roundabout Guidance Receptor 4) promoter [17]. In that case, in vivo target cell selectivity was validated in a model with disseminated cancer. Both approaches show potential for cell-targeted adenoviral therapies in the future, minimizing side effects and maximizing efficiency of the therapy.

Another point of interest is the delivery route, as it was shown that different administration routes for the adenovirus gene transfer vectors could present different levels of innate immune response. This was quantified in a clinical trial by measuring interleukin-6 levels (IL-6) [18]. The best-tolerated administration route was injection into the skin directly, as it only induced a mild IL-6 increase. Finally, other genetic alterations of the adenovirus-coding vector were proposed to reduce the increase in IL-6, (interferon gamma) γ-IFN, (tumor necrosis factor) TNF-α, and (interleukin 10) IL-10 after injection, such as the one described for the ONYX-015 adenovirus (Onyx Pharmaceuticals), an adenovirus E1B-55kD (E1B) gene-deleted replication selective adenovirus. Intravenous infusion of ONYX-015 was better tolerated at doses up to 2 × 10^13^ particles when compared to classical adenoviruses [19]. These different optimization steps show that it is possible to maximize the efficiency of adenoviral-based therapies and minimize immune inflammation using different complementary approaches.

Non-invasive imaging, such as computer tomography (CT) scans, has been described as the best approach to assess the efficiency of adenoviral-based gene therapy in lung adenocarcinoma, as indicated by a clinical trial in 1999 (Table 1). Positron emission tomography computer tomography (PET-CT) has also been used in more recent clinical trials to monitor tumor growth in lung cancer patients. Recent work on genetically engineered animal models of lung cancer also used CT or PET-CT to evaluate the efficiency of potential therapies in preclinical models. These new non-invasive approaches for the monitoring of lung cancer development in model animals have been shown to be essential for the efficient testing of adenoviral-based therapies in genetically engineered models of lung cancer. In line with these findings, modifiers of oncoproteins have recently provided a new pool of therapeutic targets for lung cancer that have only been investigated in recent years. The status of these protein modifiers, such as phosphatases, kinases, ubiquitin ligases, or hydrolases, has been often overlooked in previous clinical trials. However, these modifiers are often mutated in a significant percentage of lung cancer patients who also present classical oncogenic mutations. According to The Cancer Genome Atlas (TCGA), the most common mutations occurring in lung cancer affect *KRAS* (Kirsten rat sarcoma virus) or *EGFR* (epidermal growth factor receptor). Mutations of *BRAF* (B-Raf proto-oncogene) or *ERBB2* (v-erb-b2 avian erythroblastic leukemia viral oncogene homolog 2) occur in a smaller subset of patients. Some phosphatases or kinases associated with these pathways are mutated in up to 40% of lung cancer patients, further indicating the relevance of therapies targeting these modifiers of oncogenic signaling. Mutations of the kinases or phosphatases indicated in Table 2 often co-occur with genetic mutations of drivers of the oncogenic mutations, such as *KRAS*, confirming the rationale for the investigation of these as potential therapeutic targets.

A few examples of potential therapeutic targets regulating post-translational modifications of oncoproteins are listed in Table 2, with details including the model used, the delivery route, and the promoter used for the adenoviral construct, as various options exist when using mouse models of lung cancer. First, an adenovirus that delivered expression of mutationally activated Phosphoinositide 3-Kinase (PI3K) appeared to modulate the activity of BRAF in a murine model of lung cancer with a *BRAF* activating mutation. PI3K is essential for oncogenic signal transduction as it directly regulates downstream effectors important for transcription or metabolic deregulation. This indicates that re-activating this kinase using adenovirus-based therapy could be used to treat this type of cancer in combination with classical drugs used in lung cancer presenting *BRAF* mutations, such as BRAF or MEK inhibitors, dabrafenib, or trametinib. This could also uncover therapeutic options involving kinases that are commonly inactivated in cancer cells for lung cancer patients who are resistant to such therapies (Table 2, [20]).

Another regulator of oncogenesis in lung cancer is PP2A (protein phosphatase 2A), a phosphatase that regulates the downstream functions of KRAS. One recent study identified PP2A as a key modulator of resistance to MEK inhibition in KRAS mutant lung cancer, across a library of more than 200 kinase inhibitors. This indicates that anti-PP2A therapy could be a valid option for drug-resistant lung cancer. Furthermore, it was recently shown that KRAS G12D-induced lung tumorigenesis was significantly accelerated in Ppp2r4 gene-trapped mice as compared to Ppp2r4 wild-type [21]. This evidence was obtained through adenoviral delivery through the intratracheal route, further demonstrating the therapeutic potential of targeting PP2A, as PP2R4 encodes for the PP2A activator PTPA. Furthermore, a kinase inhibitor screen revealed that PPP2R4-depletion induced resistance against selumetinib (a MEK inhibitor), further confirming the potential of anti-PP2A therapy in MEK inhibitor-resistant lung cancer. This is further evidence of the potential of anti-adenoviral therapy with phosphatases or kinases associated with tumor transformation as combination therapy for lung cancer patients. This is a clear advance for the field in the identification of novel therapeutic targets.

The phosphatase tumor suppressor phosphatase and tensin homolog deleted on chromosome 10 (PTEN) also appears to have strong therapeutic potential for targeting through viral-based approaches in vivo. PTEN is a negative regulator of the phosphoinositide 3-kinase/AKT survival pathway, that exhibits strong tumor-suppressive activities, as the PI3K/AKT signaling pathways are activated by KRAS and EGFR mutations. Combined adenovirus-mediated PTEN (AdVPTEN) gene therapy with DNA-damaging anticancer drug cisplatin chemotherapy, appeared to dramatically reduce the progression of a murine model of lung cancer, due to induction of growth arrest and apoptosis in tumor cells. This shows that adenoviral delivery of PTEN could restore sensitivity to cisplatin in resistant lung cancers, further strengthening the hope for novel adenoviral therapies in cancer. These three examples demonstrate the potential for adenoviral-based therapies in the targeting of kinases or phosphatases specifically dysregulated in lung cancer, as this could reduce off-target effects and enable the design of better combinational therapies for patients with lung cancer.

## 3. PROteolysis TArgeting Chimeras (PROTACs) Based Therapies

Degradation of targeted proteins using proteolysis targeting chimeras (PROTACs) has recently gained a lot of momentum. A PROTAC is a bifunctional molecule that consists of different parts: a ligand that interacts with the protein to be degraded and another ligand that binds to an E3 ubiquitin ligase to promote the degradation of a specific target. The efficacy of PROTACs has been shown to be hard to predict, especially in the context of cancer. A recent article indicated that the design of the PROTACs should be optimized for each type of cancer [24], as the level of E3 ligases/degradable targets is different in each. In the case of lung cancer, different approaches were tested, as listed in Figure 1/Table 3 below. Recent clinical trials demonstrate the safety and the retainment of pharmacokinetic and therapeutic efficiency of the PROTACs in the body. The delivery of PROTACs to patients was challenging in the past, as the size of PROTACs was too important, impairing cell cellular membrane penetration. Recent optimizations for the delivery of the molecules include switching from Von Hippel Lindau (VHL) to Cereblon (CRBN)-recruiting/phthalimide-based PROTACs, which show better bioavailability even when delivered orally to patients. VHL is often mutated in Von Hippel–Lindau disease, possesses E3 ubiquitin ligase activity, and acts as an adaptor of the Cullin-2 (CUL2) RING ubiquitination complex. One of the main substrates of VHL is HIF1 (hypoxia-inducible factor 1), whose level is finely regulated through ubiquitination in every cell type. CRBN, on the other hand, is an E3 ubiquitin ligase adaptor of the CUL4A ubiquitination complex with a strong affinity for damaged DNA binding protein 1 (DDB1). Both ubiquitin adaptors are expressed at high levels and present viable options for the design of PROTACs. This shows that while PROTACs’ size can be seen as a problem for delivery, it is still possible to optimize PROTACs by reducing their size or changing their recruiter to maximize their delivery of tumor cells in patients. The PROTACs present the advantage of being able to target a specific oncogenic mutation or a specific genetic interaction between two proteins, only present in tumor cells.

Kirsten rat sarcoma virus (KRAS) mutations are responsible for up to 30% of lung cancer cases, with the *KRAS G12C* oncogenic mutation being the most frequent lung cancer driver. A recent study targeted KRAS G12C specifically in lung cancer, using a PROTAC that engaged Cereblon (CRBN). The PROTAC interacted with KRAS G12C in vitro, induced dimerization with CRBN, and degraded GFP-KRAS G12C in cells when overexpressed, but failed to degrade endogenous KRAS G12C in lung cancer cells [25]. This data suggests that while a PROTAC degrader for oncogenic KRAS could show some strong therapeutic potential, the degrader should be optimized to effectively polyubiquitinate endogenous KRAS to show any anti-tumor activity. Another more successful approach was to target KRAS G12C using a MRTX849 warhead to involve VHL. MRTX849 is a potent, orally available, and mutation-selective covalent inhibitor of KRAS G12C. In that case, this PROTAC showed a striking effect on RAS-dependent signaling, such as the MAPK cascade in cells with KRAS G12C mutations [26]. While these PROTACs showed promising effects in vitro or in xenografts, their efficiency in genetically engineered mouse models of lung cancer or in lung cancer patients still needs to be demonstrated. Furthermore, while *KRAS-G12C* mutations are the most common, other residues are often mutated, causing cancer, such as *G12D*, *G12V*, or *Q61N*. These different mutations should also be targeted to propose therapies for patients presenting these different mutations, making it a challenge as it would be essential to propose targeted therapies for each type of genetic alteration affecting *KRAS*.

Another regulator of oncogenic progression in lung cancer is son of sevenless homolog 1 (SOS1). Targeting SOS1 is of strong interest because such an approach could be used in patients with different oncogenic KRAS mutations, regardless of the residue where the mutation occurs. SOS1 is a guanine nucleotide exchange factor (GEF) that acts as a downstream regulator of KRAS signaling, for which novel inhibitors were recently developed for combination anti-cancer therapies [31]. SOS1 is a guanine nucleotide exchange factor (GEF) that interacts with KRAS to phosphorylate GDP into GTP, a step essential for KRAS oncogenic signal transduction. The first SOS1 PROTACs were described recently by connecting a VHL ligand to a previously described SOS1 agonist, which disrupt SOS1 and oncogenic KRAS protein interaction. Several PROTACs were tested, and the best ones induced SOS1 degradation in both pancreatic and lung KRAS-driven cancer cells, showing clear antiproliferative activity when compared to the SOS1 agonist itself. A tumor xenograft study further confirmed and demonstrated the antitumor potency of the SOS1-PROTACs in vivo. The demonstration of the in vivo effect of this PROTAC is promising for lung cancer patients with KRAS mutations, as chemicals targeting SOS1-based therapy are currently the subject of ongoing clinical trials [32]. This is another tool in the arsenal against KRAS-mutated lung cancer and could contribute to the design of combination therapy for patients with KRAS mutated lung cancer.

Focal adhesion kinase (FAK) was also described as a promising therapeutic target for KRAS-mutant lung cancer. However, clinical trials testing FAK inhibitors only showed limited antitumor activity [33]. Recent work used a FAK-targeting PROTAC using the FAK ligand, Tofacitinib, to degrade FAK protein via the ubiquitin-proteasome system in KRAS mutant lung cancer cells [20]. The PROTAC degraded FAK even at low doses and demonstrated striking anti-tumor activity in vitro and in a xenograft model, when compared to the anti-FAK drug tested in clinical trials, defactinib. This is further evidence of the potential for targeting kinases associated with oncogenic progression to block tumor growth and potentiate classical anti-lung cancer therapies. Combining anti-FAK and MEK therapy would be valuable in lung cancer patients presenting *KRAS* or *BRAF* mutations, as these alterations are known to activate the FAK signaling cascade on top of the classical MAPK cascade involving MEK1/2.

Another commonly mutated genetic driver in lung cancer is the epidermal growth factor receptor (EGFR), which is mutated in up to 25% of lung cancer patients. The standard of care for patients with EGFR mutations is targeted therapy with tyrosine kinase inhibitors (TKIs), but these treatments can lead to drug resistance. More than half of the drug resistance due to the EGFR TKIs was caused by an acquired T790M mutation in the kinase domain of the EGFR protein. Recent studies have described PROTACs designs that target a specific EGFR mutation using a Cereblon (CRBN)-based approach [21]. The PROTACs were designed using the EGFR inhibitor canertinib and the cereblon ligand pomalidomiden and showed anticancer activity in lung cancer cells with EGFR mutations. However, no effect was seen in lung cancer cells with wild-type EGFR, showing the specificity of the compound. Another PROTAC approach targeted EGFRL858R/T790M and EGFRdel19 using a novel covalent purine-containing EGFR ligand, which demonstrated a strong anticancer effect in cell line with EGFR mutations [22].

Finally, as an alternative, a novel technology has recently emerged to target ubiquitin hydrolases (called DUB-TACs). DUBTACs are heterobifunctional molecules consisting of a protein-targeting ligand linked to a DUB recruiter via a linker (Figure 1a). DUBTACs are ideally used to stabilize the levels of actively ubiquitinated proteins that are then degraded by the proteasome. When treating cells, a DUBTAC would target the ubiquitin hydrolases to a substrate to remove polyubiquitin chains to prevent the protein from undergoing degradation, causing stabilization of otherwise actively degraded proteins [34]. A recent publication described a covalent small-molecule recruiter for the K48-ubiquitin chain-specific DUB OTUB1 [35], which is actively involved in the stabilization of proteins associated with tumor progression such as KRAS [36]. OTUB1 is of strong interest for this approach as it has a high affinity for Lysine 48 (K48)-bound ubiquitin chains, one of the types of ubiquitination targeting proteins to the proteasome. Designing DUBTACs for OTUB1 could enable the reduction of prodegradative ubiquitination of tumor suppressors, leading to their stabilization (Figure 2b). DUBTACs is an intriguing strategy that could be used in the future for the stabilization of suppressors of oncogenic transformation, such as mutationally activated targets promoting tumor dedifferentiation. These complementary approaches reveal novel therapeutic options in lung cancer, opening the door to combination therapies.

## 4. Lipid Nanoparticle-Based Therapies

Delivery of an active drug or gene therapy to target tumor growth has been successfully achieved through specific structures, called lipid nanoparticles (LNPs), engineered to preserve the active components’ function while reducing side effects. The first attempt at cancer immunotherapies in 1995 was based on mRNA delivery via LNPs, when injection of mRNA encoding carcinoembryonic antigens elicited antigen-specific immune responses in mice. Following the wide acceptance of lipid nanoparticle-based anti-COVID-19 (Coronavirus Disease 19) vaccines [37], a number of mRNA-based cancer vaccines are currently undergoing clinical trials based on lipid nanoparticle–mRNA formulations. So far, most clinical trials have focused on delivering classical clinical drugs, such as docetaxel/paclitaxel, or PD1 (programmed cell death 1), in combination with cancer-specific antigens (FixVac-PD1) to increase the immune response [38] (Table 4). Recent studies have also targeted kinases or phosphatases involved in the oncogenic signaling cascade. While not all focusing on lung cancer, previous clinical studies on solid/epithelial tumors show that the lipid nanoparticle-based treatments are generally well tolerated with few side effects in patients, at least during the monitoring of the trials. Further investigation is required to assess the long-term effect of lipid nanoparticle delivery, especially on the cardiovascular system. It is also possible to target the particles to specific cell types using specific adjustments to the lipid nanoparticles coating the vesicles. Several examples were described, including targeting specific immune cell populations to impair tumor progression [39], tumor cells directly, or tumor associated vessels [40]. This targeting involves various modifications, such as magnetoliposomes, that enable specific cell targeting of specific cell types according to their magnetic properties. Targeting the lipid nanoparticles to specific cell types would reduce any side effects to a minimaum, enabling the safest setup for such therapies. As described in Table 4, some clinical trials involving nanoparticles reached Phase III and IV, showing the acceptance of this strategy in the targeting of cancer progression. These phase III and IV trials included different targets, including the PD1 receptor, paclitaxel, or A-type CpG-oligonucleotides specific to cancer cells (Table 4, [41,42,43]).

For instance, lipid nanoparticles encasing mRNA encoding for Polo-like kinase1 (PLK1), kinesin spindle protein (KSP), and centromeric protein E (CENPE) have been recently described [56]. Targeting these kinases induces mitotic arrest and blocks tumor progression in different models, including lung cancer. Other studies directly target proteins involved in oncogenic signaling driven by specific genetic mutations, such as BAX, to activate the pro-apoptotic BCL2 family in cancer cells [57]. Finally, the phosphatase PTEN involved in PI3K-AKT signaling activated by RAS signaling, could also be targeted by nanoparticles. A recent paper gives a clear demonstration of the potential of reactivation of PTEN using mRNA nanoparticles to enhance antitumor immunity in different types of cancer [58]. This publication is a clear demonstration of the potential of anti-kinase or anti-phosphatase therapy for the treatment of cancer using mRNA-based liposomal formulations. A formulation of LNP delivering siRNAs targeting VEGF and kinesin spindle protein (KSP) also showed efficacy in patients with liver lesions from diverse tumor types [59].

Beyond LNP or AAV delivery of oligonucleotides, other strategies have proved successful. Systemic administration of the antisense oligonucleotide (ASO) targeting BCL2, augmerosen, downregulated the target BCL2 protein in metastatic cancer and, combined with standard anticancer therapy, helped resensitize patients with resistant melanoma [60]. Similarly, the treatment with the ASO, AZD9150, decreased STAT3 expression in several preclinical models and proved to have an antitumor activity in lymphoma and lung cancer models [61]. These recent advances in the field of lipid nanoparticle-based drug delivery further demonstrate the strong potential of such approaches for the treatment of drug resistant lung cancer or the development of novel combination therapies. This strategy based on nanoparticles shows strong potential for targeted therapy in cancer patients, as it would make it possible to treat cancer cells specifically without affecting the other tissues and maximize the anti-cancer effect with low doses.

In conclusion, we describe several therapeutic approaches that focus on protein modifications that are dysregulated in the progression of lung or epithelial derived cancer. Protein modifications include phosphorylation, or the addition of a phosphorylated residue to a substrate, and ubiquitination, the addition of a ubiquitin molecule that induces targeting to the proteasome of the vesicular system. While ubiquitination is involved in protein degradation through the proteasome and trafficking to vesicles, phosphorylation is important for the (in)activation of proteins, especially in signaling cascades such as the mitogen activated protein kinase (MAPK) signaling cascade that is involved in cancer cell proliferation and survival. Kinases and phosphatases are enzymes involved in the phosphorylation or removal of the phosphate molecule(S) from a cargo, respectively. Several therapeutic design options focusing on specific kinases or phosphatases dysregulated in lung cancer progression were developed in the past years with various levels of success. These therapeutic designs included adenoviral-based or lipid nanoparticle-based strategies, showing good bioavailability and tolerance in patients. Another approach exploiting the ubiquitin framework is the PROTACs or DUBTACs strategy, which also showed proper response in patients, after optimization of the delivery of the compound, which was the most limiting factor in that case.

Our comprehensive integration of clinical trials and recent findings in fundamental research in the field properly demonstrates the strong interest for these novel pharmaceutical strategies. While the adenoviral-based therapies are the oldest and most tested ones, they appear to be less well tolerated when compared to mRNA-based lipid nanoparticle delivery systems, according to the different clinical trials. The mRNA-based lipid nanoparticle drugs showed fewer side effects in the different trials and are now widely accepted after the massive use of mRNA vaccines for COVID-19. This explains a shift towards lipid nanoparticles recently. However, adenoviral-based approaches can still be optimized through genetic engineering, such as the use of targeted promoters for cell-specific expression. While genetic targeting using adenoviruses or lipid nanoparticles is precise, modulating specific protein modifications occurring specifically in cancer cells could be even better tolerated. PROTACs are one of the key technologies for the degradation of oncoproteins and show tremendous potential. However, this technology is still much less advanced when compared to lipid nanoparticle-based mRNA delivery or adenovirus based therapy, as indicated by the comparatively small number of clinical trials ongoing with this technology [62].

## Figures and Tables

**Figure 1 pharmaceutics-15-00206-f001:**
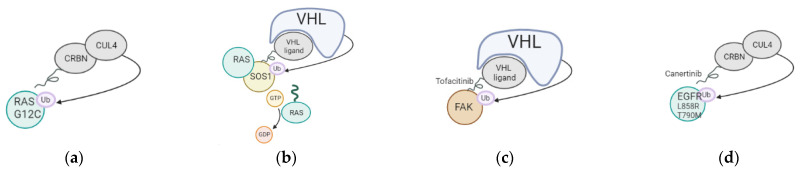
Mechanism of action of PROTACs tested in lung cancer. (**a**) Targeting of KRAS G12C oncogenic mutation using a Cereblon-based PROTAC; (**b**) Targeting of SOS1, cofactor of KRAS, using a VHL-based PROTAC; (**c**) Targeting of FAK using a VHL-based PROTAC; (**d**) Targeting of EGFR L858R and T790M using a CRBN-based PROTAC.

**Figure 2 pharmaceutics-15-00206-f002:**
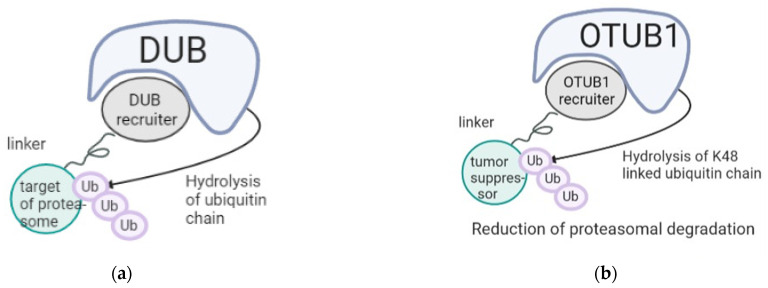
DUBTACs as promising anti-cancer approaches. (**a**) Mechanism of action of DUBTACs that hydrolyze pro-degradative ubiquitin chains attached to a target to reduce recruitment of the cargo protein to the proteasome, thus reducing its degradation. (**b**) Hypothetical design using DUBTACs to recruit OTUB1 as a strategy to stabilize tumor suppressors through deubiquitination of K48-bound ubiquitin chains.

**Table 1 pharmaceutics-15-00206-t001:** Clinical trials using adenovirus-based therapies in lung cancer.

Clinical Trial Name	Drug Tested	Status	Ref
Phase I study of recombinant adenovirus-mediated gene transfer in lung cancer patients.	replication-defective adenoviral expression vector encoding wild-type p53	Completed	[1]
A phase I study of adenovirus-mediated wild-type p53 gene transfer in patients with advanced non-small cell lung cancer.	replication-defective adenoviral expression vector encoding wild-type p53	Completed	[2]
Phase I trial of sequential administration of recombinant DNA and adenovirus expressing L523S protein in early-stage non-small-cell lung cancer.	L523S, an immunogenic lung cancer antigen delivered into an E1B-deleted adenovirus (Ad/L523S)	Completed	[11]
Phase II trial using dendritic cells transduced with an adenoviral vector containing the p53 gene to immunize patients with lung cancer	Efficacy of paclitaxel following the dendritic cell (DC)-based p53 vaccine (Ad.p53-DC vaccine),	Completed	[3]
Adenovirus-mediated wild-type p53 gene transfer in combination with bronchial arterial infusion for treatment of advanced non-small-cell lung cancer, one year follow-up	rAd-p53 and BAI	Completed	[4]
CT-guided intratumoral gene therapy in non-small-cell lung cancer.	replication-defective adenoviral expression vector encoding wild-type p53	Completed	[5]
Gene therapy plus radiation therapy in treating patients with non-small cell lung cancer	adenovirus p53 gene therapy and radiotherapy	Completed	[6]
Phase I study of adenovirus p53 administered by bronchoalveolar lavage in patients with bronchioloalveolar cell lung carcinoma	replication-defective adenoviral expression vector encoding wild-type p53	Completed	[7]
Chemotherapy followed by vaccine therapy in treating patients with extensive-stage small cell lung cancer	Dendritic cell (DC)-based p53 vaccine and trans-retinoic acid	Completed	[8]
Safety and efficacy of recombinant oncolytic sdenovirus L-IFN injection in relapsed/refractory solid tumors clinical study (YSCH-01)	Ad5-yCD/mutTKSR39rep-ADP adenovirus, 5-fluorocytosine (5-FC), valganciclovir (vGCV), 48Gy	Oncolytic effect, withdrawn	[9]
TNFα and IL-2 coding oncolytic adenovirus TILT-123 monotherapy (TUNIMO)	pVAX/L523S and two doses of Ad/L523S	T-cell activating, status unknown	[12]
Oncolytic MG1-MAGEA3 with Ad-MAGEA3 vaccine in combination with pembrolizumab for non-small cell lung cancer patients	Ad-MAGEA3 and MG1-MAGEA3 in combination with pembrolizumab	T-cell activating, status unknown	[13]
Safety study of human MUC-1 (Mucin-1) adenoviral vector vaccine for immunotherapy of epithelial cancers (MUC-1)	Ad-sig-hMUC-1/ecdCD40L vector encoding a fusion protein in which hMUC-1 epithelial antigen is attached to CD40L (CD40 ligand)	Dendritic cell activating, status unknown	[10]

**Table 2 pharmaceutics-15-00206-t002:** Oncogenic phosphatases/kinases that could be targeted using adenoviruses.

Target of the Adenovirus	Oncogene Affected	Model Used	Delivery Route	Adenovirus Promoter	Refs.
Mutationally activated kinase PI3K (H1047R)	BRAF V600E signaling	BRAF V600E-driven lung adenocarcinoma tumors	Intratracheal injection	SFTPC Adenovirus	[20]
Phosphatase PP2A	KRAS G12D signaling	KRAS G12D induced lung adenocarcinoma tumors	Intratracheal injection	Constitutive Adenovirus	[21,22]
Phosphatase PTEN	KRAS signalling	Xenograft (NCI-H446 lung small cell cancer cell)	Intratumor injection	Constitutive adenovirus	[23]

**Table 3 pharmaceutics-15-00206-t003:** PROTACs tested in lung cancer with details of degrader and recruited ligase.

Target	Degrader	E3 Ligase	References
KRAS G12C (a)	Cereblon CRBN	CUL4	[25,26]
SOS1 (b)	SOS agonist	VHL	[27]
FAK (c)	Tofacitinib (FAK ligand)	VHL	[28]
EGFR L858R+T790M (d)	Cereblon CRBN	CUL4	[29]
EGFR	novel covalent purine-containing EGFR ligand		[30]

**Table 4 pharmaceutics-15-00206-t004:** Clinical trials with lipid nanoparticles in solid tumors.

Study Type	Target	Status	Reference
Phase II	BIND-014 (docetaxel nanoparticles) in lung cancer	Completed	[44]
Phase II	BIND-014 (docetaxel nanoparticles) in KRAS-mutated lung cancer	Completed	[45]
Phase II	Carboplatin and paclitaxel albumin-stabilized nanoparticle (ovarian)	Completed	[46]
Phase II	Paclitaxel albumin-stabilized nanoparticle in lung cancer	Completed	[47]
Phase I-II	ABI-007 (albumin-stabilised paclitaxel nanoparticle) in lung cancer	Completed	[48]
Phase II	CRLX101 (cyclodextrin-based polymer with camptothecin) in lung	Completed	[49]
Phase IV	Paclitaxel liposome in lung cancer	Completed	[41]
Phase III-IV	A-type CpG-oligonucleotides (CpG-ODN) coupled to peptide derived from Melan-A/MART-1 in melanoma	Completed	[42]
Phase III	Paclitaxel micelles in lung cancer	In progress	[50]
Phase I-II	GPX-001 (TUSC2 encapsulate in lipid nanoparticles) in lung cancer	In progress	[51]
Phase II	LY01610 (irinotecan hydrochloride liposome) in lung cancer	In progress	[52]
Phase II-III	ONIVYDE (irinotecan liposome) in lung cancer	In progress	[53]
Phase III	HLX10 (humanized—PD-1 receptor) liposome in lung cancer	In progress	[43]
Phase I-II	CRLX101 (cyclodextrin polymer with camptothecin) in lung cancer	In progress	[54]
Phase II	ABI-009 (albumin-bound mTOR inhibitor) in neuroendocrine tumor	Terminated	[55]

## Data Availability

Not applicable.

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
