# Peer review of "Novel Therapeutic Approaches Targeting Post-Translational Modifications in Lung Cancer"

_pharmaceutics, 2023, doi:10.3390/pharmaceutics15010206_

Round 1

Reviewer 1 Report

In this review paper, the authors describe recent advances of drug-delivery approaches including adenovirus, lipid nanoparticles and PROTACs, targeting post-translational modifications of proteins involved in tumorigenesis of lung cancer. It is well organized and finely polished for publication.

Author Response

We thank the reviewer for the feedback.

Reviewer 2 Report

In the review, authors described recent advances in the pharmaceutical research field for development of different technologies targeting post-translational modifications of proteins involved in tumorigenesis of lung cancer., such as adenovirus, lipid nanoparticles and PROTACs. However, I think the topic is not generally interest to readers from wide field. And the content of the review is not rich enough. Also, the review did not well organized. Therefore, This manuscript is not suitable for publication in the current state.

1.In order to let the reader better understand the topic of this review, it is recommended to add the scheme and the research purpose of this review in the introduction. Also, the background information of the topic is too little and should be supplemented.

2. The contents of all the tables in the text are too simple, especially Table 2, which is unnecessary to list the title of the article. In addition, the contents of Table 2 are repeated with the contents of the three subsequent paragraphs, which is inappropriate.

3. The descriptions of Figure 1 and Figure 2 are too brief and the mechanism of action of PROTACs and DUBTACs is not clearly expressed in the pictures, which should be redrawn.

4. Too much of contents in this review are grammatically incorrect and tense chaotic, and author needs to check and correct them carefully.

For examples:

Line 59: Adenoviruses are The efficiency of such therapies has been demonstrated in several genetic diseases (i.e. Zolgensma) and, recently, by the use of 60 adenoviral based vaccines against the COVID19 respiratory diseases.

Line 86: It is worth noting that the different adenoviral based- clinical trials indicated above in Table 1 (references 1- 10) showed side effects, including included fever, hypoxia, and dyspnea.

Line 97: Another option would be to target tumor angiogenesis to block neo-vessel formation and impair tumor progression. In that case, studies have shown that it is possible to target proliferative vascular endothelium using the ROBO4 (Roundabout Guidance Receptor 4) promoter17. In that case, in vivo target cell selectivity was validated in a model with disseminated cancer. B

Line 157: The phosphatase, Tumor suppressor phosphatase and tensin homolog deleted on chromosome 10 (PTEN) also appears to have strong therapeutic potential for targeting through viral based approaches in vivo.

Line 184: The PROTACs present the advantage ob being able to target a specific oncogenic mutation or a specific genetic interaction between two proteins, only present in tumor cells.

Line 239: However, no effect in lung cancer cells with wildtype EGFR, showing the specificity of the compound.

5. In section 3, authors would like to indicate that the vector used to deliver the gene is lipid-nanoparticles (LNP), not nanolipid.

Author Response

We thank the reviewer for the comments

We have now expanded the introduction and redesigned the graphical abstract to give a better overview of what we want to achieve with this review.

We have now revised Table 2 with more technical information and removing the titles of the articles. We hope that this new version of Table 2 is now more informative.

Figure 1 and 2 have been redrawn to explain better the mechanism of action of PROTACs and DUBTACs. The figure legends have also been expanded.

Reviewer 3 Report

Dear Authors,

I have read with interest your nice review about novel targets for cancer therapy based on post-translational modifications.

Here are some suggestions to improve on the text, especially introducing explanatory paragraphs at some key points:

1. Please, revise the text for copy and paste and typing errors. 

2. Line 127: TCGA: please explain the acronym.

3. Line 138: PI3K. Please, explain better what happens when this kinase is inactivated in cancer.

4. Line 172: Please, mention and discuss briefly, including some citations, E3 ubiquitin ligases. I would suggest to give some details at least of those present in Table 3.

5. Line 213: Please, I would better explain the role of the SOS1 agonist.

6. Line 237. Please, explain the CRBN acronym.

7. Line 248: "...causing stabilization of actively...". I would change it into "...causing stabilization of otherwise actively...".

8. Line 281: Placlitaxel should be Paclitaxel.

9. Table 4: I would better separate rows to favour clarity.

Author Response

Here are some suggestions to improve on the text, especially introducing explanatory paragraphs at some key points:

  1. Please, revise the text for copy and paste and typing errors. 
  2. Line 127: TCGA: please explain the acronym.
  3. Line 138: PI3K. Please, explain better what happens when this kinase is inactivated in cancer.
  4. Line 172: Please, mention and discuss briefly, including some citations, E3 ubiquitin ligases. I would suggest to give some details at least of those present in Table 3.
  5. Line 213: Please, I would better explain the role of the SOS1 agonist.
  6. Line 237. Please, explain the CRBN acronym.
  7. Line 248: "...causing stabilization of actively...". I would change it into "...causing stabilization of otherwise actively...".
  8. Line 281: Placlitaxel should be Paclitaxel.

We thank the reviewer for these suggestions. We have now corrected the different points.

Reviewer 4 Report

I found the manuscript " Novel therapeutic approaches targeting post-translational mod ifications in lung cancer" by Maria et al. interesting, which review describes recent advances in the pharmaceutical research field for development of these different technologies targeting post-trans lational modifications of proteins involved in tumorigenesis of lung cancer. The topic is of current interest and suited for the journal; anyways, some modifications of the submitted paper are recommended before publication.

Comments and remarks:

Abstract is written well and structured but authors may revise the objective of this review.

Quality of graphical abstract is not up too mark,

Authors may re draw the graphical abstract

Introduction:

In introduction section from line no 27-50

Authors should revise these line with new references.

Authors may cite bellow mentioned reference

https://doi.org/10.1016/j.semcancer.2021.05.018

over all manuscript is written well accept figures

figures are wel structured, authors may revise figure no. 2

Author Response

Abstract is written well and structured but authors may revise the objective of this review. 

Quality of graphical abstract is not up too mark, 

Authors may re draw the graphical abstract

We thank the reviewer for the comments and redesigned the graphical abstract to give a better overview of what the objective of this review is.

Reviewer 5 Report

Reviewer Report.

pharmaceutics-2125064

This paper presents a review related with lung cancer. It describes recent advances in the pharmaceutical research field for development of different technologies. The manuscript is well written and the three sections in which the paper is structured are well selected. However, a personal opinion of the authors is missing in the sections and especially in the final conclusions. The manuscript cannot focus only on the development and description of techniques based on on protein modifications that are dysregulated in the progression of lung or epithelial derived cancer. It is necessary, at least in the final conclusions, some lines that show future prospects where the authors clearly express their personal opinions showing their interest in writing the review and their experience. Other minor modifications:

1)     In my opinion the introduction is too short and some references must be included. Most of the comments in this section are known by people interested in the paper but I think it is necessary.

2)     Some references should be modified by more recent ones.

Author Response

However, a personal opinion of the authors is missing in the sections and especially in the final conclusions. The manuscript cannot focus only on the development and description of techniques based on on protein modifications that are dysregulated in the progression of lung or epithelial derived cancer. It is necessary, at least in the final conclusions, some lines that show future prospects where the authors clearly express their personal opinions showing their interest in writing the review and their experience.

We thank the reviewer for the comments and have now expanded the discussion with a more personal statement.

In my opinion the introduction is too short and some references must be included. Most of the comments in this section are known by people interested in the paper but I think it is necessary.

We thank the reviewer for the comments and have now expanded the introduction.

2)     Some references should be modified by more recent ones.

We decided to focus on lung cancer for the sake of clarity. We agree that additional references could be added if we also included other cancer types. We tried to give a broad overview of advances in the field of lung cancer, including older references to explain the progression in the design of therapies and what was learned in past trials.

Round 2

Reviewer 2 Report

The author has carefully revised this manuscript according to the review comments. In the current state, this manuscript can be accepted for publication.